# FEATURIZED BIDIRECTIONAL GAN: ADVERSARIAL DEFENSE VIA ADVERSARIALLY LEARNED SEMANTIC INFERENCE

## ABSTRACT

Deep neural networks have been demonstrated to be vulnerable to adversarial attacks, where small perturbations intentionally added to the original inputs can fool the classifier. In this paper, we propose a defense method, Featurized Bidirectional Generative Adversarial Networks (FBGAN), to extract the semantic features of the input and filter the non-semantic perturbation. FBGAN is pre-trained on the clean dataset in an unsupervised manner, adversarially learning a bidirectional mapping between the high-dimensional data space and the low-dimensional semantic space; also mutual information is applied to disentangle the semantically meaningful features. After the bidirectional mapping, the adversarial data can be reconstructed to denoised data, which could be fed into any pre-trained classifier. We empirically show the quality of reconstruction images and the effectiveness of defense.

## 1 INTRODUCTION

The existence of adversarial examples causes serious security concern about reliability of deep neural networks (DNN). DNN may mislabel the perturbed images with high confidence even though the perturbation is too small to be recognized by human. Moreover, adversarial examples will often fool several models simultaneously, even if these models have different architectures (Szegedy et al., 2014). One possible explanation is that when recognizing images, human usually catch high-level and semantic features, such as the shape of the digits in MNIST dataset, which are robust under small perturbation; DNN may easily catch low-level and weak features, such as the gray-scale values of certain area in the images, which are non-robust when the pixel-wise perturbation accumulates (Tsipras et al., 2018).

Most previous adversarial defense methods fall into two classes: adversarial training and gradient masking. Adversarial training methods (Szegedy et al., 2014; Tramèr et al., 2017; Madry et al., 2017; Sinha et al., 2017) apply adversarial perturbations on training data online, and feed both the clean data and the adversarial data to train the classifier, i.e., solve a minimax game iteratively. However, it is flawed by the high computational cost to generate adversarial examples, especially for more complex dataset and harder attacks. Gradient masking methods modify the architecture of the classifier such that the attacker cannot get useful gradient information of the inputs. One example is the thermometer encoding (Buckman et al., 2018) which preprocesses the input in a one hot vector, and such discretization prevent the attacker from backpropagating through the input to calculate the adversarial purtabation. However, Athalye et al. (2018) shows that gradient masking methods can be circumvented and lead to a false sense of security in defenses against adversarial attacks.

Both of adversarial training and gradient masking methods defend adversarial attacks by improving the classifier. We take another approach by denoising the adversarial examples without changing the classifier (Meng & Chen, 2017; Ilyas et al., 2017; Liao et al., 2018). Our defense is motivated by human cognition process. The fact that adversarial examples cannot fool human suggests that human do classification based on some semantic features that are unchanged after the perturbation. Hence, it is natural to extract those semantic features and doing the inference solely based on semantic information. One closely related work is Defense-GAN (Samangouei et al., 2018), which trains

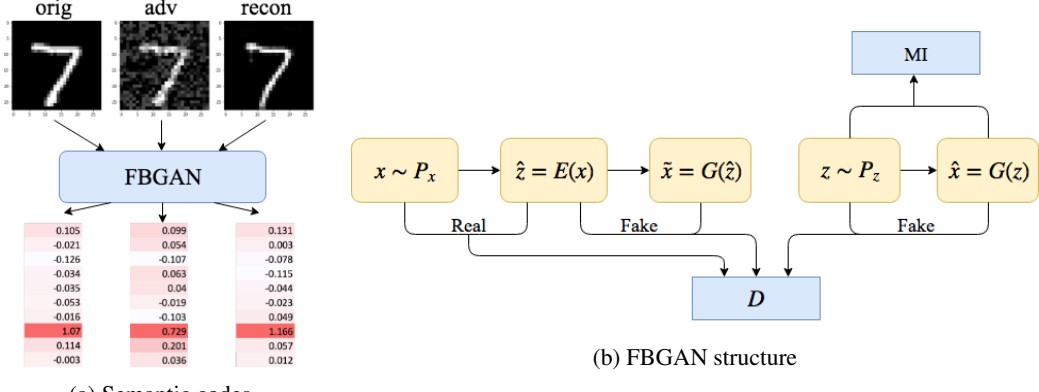

(a) Semantic codes

(b) FBGAN structure

Figure 1: (a) The semantic features of images should be unchanged before and after the adversarial perturbation. Via FBGAN, original, adversarial and reconstructed images are encoded to similar semantic codes. Each column stands for the ten-categorical code that related to the classification of an image (see section 3 for details). Here all three images are classified as "7" from categorical codes. (b) Besides a discriminator $D$ and a generator $G$ in the vanilla GAN, we add an encoder $E$ mapping from the data space to the latent space, and the discriminator $D$ takes a tuple $(\boldsymbol{x}, \boldsymbol{z})$ as input. There are three types of tuple $(\boldsymbol{x}, \boldsymbol{z})$: $(\mathbf{x}, E(\mathbf{x}))$ for $\mathbf{x} \sim P_{\mathbf{x}}$, $(G(\mathbf{z}), \mathbf{z})$ for $\mathbf{z} \sim P_{\mathbf{z}}$ and $(G(E(\mathbf{x})), E(\mathbf{x}))$ for $\mathbf{x} \sim P_{\mathbf{x}}$; the discriminator $D$ treats the first type as real and the other two as fake. Mutual information between latent codes $z$ and generated $G(z)$ is maximized in order to disentangle the semantic features.

a GAN (Goodfellow et al., 2014a) to generate the manifold of unperturbed images, then finds the nearest point on the manifold to the adversarial example as the denoising result. While it is a novel way to leverage generative model to filter the adversarial perturbation, it takes iterations to search the nearest point on the manifold, which is time consuming.

In this paper, we propose Featurized Bidirectional GAN (FBGAN), an encoding and generative model that extracts the semantic features of the input images (either original or perturbed), and reconstructs the unperturbed images from these features. We take advantage of the generative capability of Bidirectional GAN (Donahue et al., 2016; Dumoulin et al., 2016), where an encoder is learned to map the input to its latent codes directly, instead of doing the manifold search iterations. Inspired by InfoGAN (Chen et al., 2016), we maximize the mutual information (MI) between all the latent codes and the generated images. The MI regularization can significantly reduce the dimension of latent space, as well as disentangle the semantic features of inputs in different components of the latent codes, e.g., the tilt angle and stroke thickness of digits in MNIST. We call the MI-enhanced latent codes as *semantic codes* (Figure 1). FBGAN is pre-trained on the clean dataset in an unsupervised manner. With the feature-extraction and reconstruction procedure, we can denoise the adversarial examples and fed them into any pre-trained classifier, which shows effective defense against both white-box and gray-box attacks (see section 4 for details).

**Our contribution**

1. FBGAN depicts a bidirectional mapping between a high-dimensional data space and a low-dimensional semantic latent space. We can extract the semantic features of the images, which is unchanged after the adversarial perturbation; we can also generate new images with indicated semantic features, such as the category and tilt angle of the digits.

2. We denoise the adversarial example by extracting semantic features and reconstructing via FBGAN. This defense method is shown to be effective for any given pre-trained classifier under both white-box and gray-box attacks.

## 2 PRELIMINARIES

### 2.1 GENERATIVE ADVERSARIAL NETWORKS AND ITS DERIVATIVES

**Generative Adversarial Networks**  GAN (Goodfellow et al., 2014a) is a generative model to learn high-dimensional data distribution via an adversarial process. Instead of modeling the probability density function, GAN learns a generator $G$ which is a mapping from low-dimensional latent space $\Omega_{\boldsymbol{z}}$ to high-dimensional data space $\Omega_{\boldsymbol{x}}$. Then a standard distribution (usually Gaussian) $\mathbf{z} \sim P_{\mathbf{z}}$ in the latent space can be transferred into the distribution $G(\mathbf{z}) \sim P_G$ in the data space. $P_G$ is supposed to approximate the objective data distribution $P_{\mathbf{x}}$, thus a discriminator $D$ is proposed to distinguish between samples from $P_{\mathbf{x}}$ and $P_G$. The generator $G$ and discriminator $D$ are represented by DNN and updated in the following minimax game:

$$\min_G \max_D V_{\text{GAN}}(D, G) := \mathbb{E}_{\mathbf{x} \sim P_{\mathbf{x}}}[\log D(\mathbf{x})] + \mathbb{E}_{\mathbf{z} \sim P_{\mathbf{z}}}[\log(1 - D(G(\mathbf{z})))]. \quad (1)$$

It can be shown that the theoretical optimal discriminator $D^\star$ satisfies:

$$D^\star(\boldsymbol{x}) = \frac{P_{\mathbf{x}}(\boldsymbol{x})}{P_{\mathbf{x}}(\boldsymbol{x}) + P_G(\boldsymbol{x})}, \quad V_{\text{GAN}}(D^\star, G) = 2D_{\text{JS}}(P_{\mathbf{x}} \| P_G) - 2 \log 2, \quad (2)$$

where $P(\cdot)$ denotes the probability density of distribution $P$, and $D_{\text{JS}}$ is the Jensen-Shannnon divergence between two distribution. Thus the theoretical optimal generator $G^\star$ will recover the data distribution, i.e. $P_{G^\star} = P_{\mathbf{x}}$.

**Bidirectional GAN**  BiGAN (Donahue et al., 2016; Dumoulin et al., 2016) considers the inverse mapping of the generator to learn the latent codes $\boldsymbol{z}$ as feature representation given data $\boldsymbol{x}$. The encoder $E$ is introduced as a mapping from data space $\Omega_{\boldsymbol{x}}$ to latent space $\Omega_{\boldsymbol{z}}$, and the discriminator takes a tuple of data point and latent codes $(\boldsymbol{x}, \boldsymbol{z})$ as inputs, distinguishing between the joint distribution of $(\mathbf{x}, E(\mathbf{x}))$ and $(G(\mathbf{z}), \mathbf{z})$. The minimax objective becomes

$$\min_{G,E} \max_D V_{\text{BiGAN}}(D, G, E) := \mathbb{E}_{\mathbf{x} \sim P_{\mathbf{x}}}[\log D(\mathbf{x}, E(\mathbf{x}))] + \mathbb{E}_{\mathbf{z} \sim P_{\mathbf{z}}}[\log(1 - D(G(\mathbf{z}), \mathbf{z}))]. \quad (3)$$

The optimal condition for $D^\star$ is replacing $P_{\mathbf{x}}$ and $P_G$ by $P_{\mathbf{x}, E(\mathbf{x})}$ and $P_{G(\mathbf{z}), \mathbf{z}}$ in (2). The optimal encoder and generator can guarantee $G^\star(E^\star(\boldsymbol{x})) = \boldsymbol{x}$ for $\boldsymbol{x} \in \Omega_{\boldsymbol{x}}$ and $E^\star(G^\star(\boldsymbol{z})) = \boldsymbol{z}$ for $\boldsymbol{z} \in \Omega_{\boldsymbol{z}}$.

**InfoGAN**  InfoGAN (Chen et al., 2016) is an extension of GAN that is able to learn disentangled semantic representation. For example, one discrete latent code may represent the class of the image while another continuous code may control tilt angles. InfoGAN decomposes the latent codes into two parts $\boldsymbol{z} = (\boldsymbol{c}, \boldsymbol{z}')$ where the semantic codes $\boldsymbol{c}$ target the meaningful features, and noise codes $\boldsymbol{z}'$ which stand for incompressible noise. Then an information-theoretic regularization is introduced to maximize MI between semantic codes $\mathbf{c}$ and generated $G(\mathbf{c}, \mathbf{z}')$:

$$\min_G \max_D V_{\text{InfoGAN}}(D, G) := \mathbb{E}_{\mathbf{x} \sim P_{\mathbf{x}}}[\log D(\mathbf{x})] + \mathbb{E}_{\mathbf{z} \sim P_{\mathbf{z}}}[\log(1 - D(G(\mathbf{z})))] - \lambda I(\mathbf{c}; G(\mathbf{c}, \mathbf{z}')), \quad (4)$$

where the mutual information $I(\mathbf{c}; \mathbf{x}) = H(\mathbf{c}) - H(\mathbf{c}|\mathbf{x})$ and $H$ is the entropy.

### 2.2 ADVERSARIAL ATTACKS

In the image classification task, given a vectorized clean image $\boldsymbol{x} \in [0, 1]^d$, a classifier $C$ will output a label $y = C(\boldsymbol{x})$. All adversarial attacks aim to find a small perturbation $\boldsymbol{\rho}$ to fool the classifier such that $C(\boldsymbol{x} + \boldsymbol{\rho}) \neq y$ (Szegedy et al., 2014). It can be formulated as

$$\min_{\boldsymbol{\rho}} \|\boldsymbol{\rho}\|, \quad \text{s.t. } \boldsymbol{x} + \boldsymbol{\rho} \in [0, 1]^d, \ C(\boldsymbol{x} + \boldsymbol{\rho}) \neq y.$$

Various attacking algorithms have been proposed to fool DNN (Akhtar & Mian, 2018; Papernot et al., 2016), and here are two most famous attacks.

**Fast Gradient Sign Method**  FGSM (Goodfellow et al., 2014b) is a single-step attack. Let $L(\boldsymbol{x}, y)$ be the loss function of the classifier $C$ given input $\boldsymbol{x}$ and label $y$. FGSM defines the perturbation $\boldsymbol{\rho}$ as

$$\boldsymbol{\rho} = \varepsilon \cdot \text{sign}(\nabla_{\boldsymbol{x}} L(\boldsymbol{x}, y)),$$

where $\varepsilon$ is a small scalar. FGSM simply chooses the sign of change at each pixel to increase the loss $L(\boldsymbol{x}, y)$ and fool the classifier.

**Projected Gradient Descent** PGD (Madry et al., 2017) is a more powerful multi-step attack with projected gradient descent:

$$x_0^{\text{PGD}} = x, \quad x_{t+1}^{\text{PGD}} = \Pi_{\mathcal{S}} \left[ x_t^{\text{PGD}} + \alpha \cdot \text{sign} \left( \nabla_x L(x_t^{\text{PGD}}, y) \right) \right]$$

where $\Pi_{\mathcal{S}}$ is the projection onto $\mathcal{S} = \{ x' : \| x' - x \|_\infty \le \varepsilon \}$.

# 3 FEATURIZED BIDIRECTIONAL GAN

## 3.1 ROUTE MAP

We use BiGAN framework to adversarially learn the bidirectional feature mapping, and MI regularization to reduce the dimension of semantic codes and disentangle the semantic features. In adversarial defense task, first we train FBGAN on clean dataset, which is an unsupervised learning for semantic encoder $E$ and image generator $G$. Second, given a pre-trained classifier $C$ and adversarial data $x$, we reconstruct $x$ as $\tilde{x} = G(E(x))$ to filter the non-semantic noise, then feed $\tilde{x}$ to the classifier and use $C(\tilde{x})$ as the prediction.

## 3.2 FORMULATION

BiGAN provides a good approach to map high-dimensional image data $x$ to low-dimensional latent codes $z = E(x)$, yet it has no restriction on the semantic meaning of the latent codes $z$. To eliminate the non-semantic noise in adversarial examples, we maximize mutual information between latent codes $z$ and generated $G(z)$. Unlike InfoGAN where the latent codes is decomposed into semantic codes and incompressible noise $z = (c, z')$ and only $I(c; G(c, z'))$ is maximized, here we regard all latent codes as semantic and maximize $I(z, G(z))$ directly. Although the former method may improve the diversity of the generation, our method focuses on the main semantic features which is more robust under adversarial attack.

To maximize the mutual information $I(\mathbf{z}; G(\mathbf{z}))$, we use Variational Information Maximization technique. Suppose the underlying joint distribution is $(\mathbf{x}, \mathbf{z}) \sim P$, then

$$I(\mathbf{z}; \mathbf{x}) = H(\mathbf{z}) - H(\mathbf{z}|\mathbf{x}) = H(\mathbf{z}) + \mathbb{E}_P[\log P(\mathbf{z}|\mathbf{x})] = H(\mathbf{z}) + \max_Q \mathbb{E}_P[\log Q(\mathbf{z}|\mathbf{x})],$$

where $Q$ is taken over all possible joint distributions of $(\mathbf{x}, \mathbf{z})$. Assume that each semantic codes $z$ contain one categorical code $z_c$ and $n$ continuous codes $z_1, \ldots, z_n$. Assume that $Q(\cdot|x)$ is a factored distribution $Q(z|x) = Q_c(z_c|x) \prod_{i=1}^n Q_i(z_i|x)$. For the categorical code, rewrite the discrete probability $Q_c(\cdot|x)$ as a vector $\varphi_c(x)$, i.e. $\varphi_c(x)_k = Q_c(z_c = k|x)$, then $\log Q_c(z_c|x) = -H(z_c, \varphi_c(x))$ where $H$ is the cross entropy of two vectors regarding $z_c$ as a one-hot vector. For the continuous codes, assume $Q_i(\cdot|x)$ is a Gaussian $\mathcal{N}(\varphi_i(x), \sigma^2)$ for fixed variance $\sigma$. Now, define *MI gap* as the following distance

$$\text{dist}(z, \varphi(x)) := -\log Q(z|x) = H(z_c, \varphi_c(x)) + C \sum_{i=1}^n \| z_i - \varphi_i(x) \|^2 \tag{5}$$

where $\varphi$ is the concatenation of $(\varphi_c, \varphi_1, \ldots, \varphi_n)$ and $C$ is a constant. Note that the MI gap is a useful approach to maximize MI between two variables.

In the defense task, we want to pay more attention to the encoding $E(x)$ and reconstruction $G(E(x))$ on given data $x$, and take the pair $(G(E(x)), E(x))$ into consideration. Therefore, FBGAN has the following objective function (as illustrated in Figure 1)

$$\min_{G, E, \varphi} \max_D V_{\text{FBGAN}}(D, G, E) := \mathbb{E}_\mathbf{x} \left[ \log D(\mathbf{x}, E(\mathbf{x})) \right]$$

$$+ \frac{1}{2} \left[ \mathbb{E}_\mathbf{z} \left[ \log(1 - D(G(\mathbf{z}), \mathbf{z})) \right] + \mathbb{E}_\mathbf{x} \left[ \log(1 - D(G(E(\mathbf{x})), E(\mathbf{x}))) \right] \right] + \lambda \mathbb{E}_\mathbf{z} \, \text{dist}(\mathbf{z}, \varphi(G(\mathbf{z}))). \tag{6}$$

## 3.3 IMPLEMENTATION

Figure 2 shows the implementation of FBGAN. $E$, $G$ and $D$ take the standard BiGAN architectures (Dumoulin et al., 2016). We replace all ReLU activation with ELU in $E$ and $G$ for smoothness, and use weight normalization instead of batch normalization in order to ensure $E(x)$ and $G(z)$ depend only on $x$ and $z$ instead of the whole minibatch (Kumar et al., 2017). $E$ are trained by feature matching methods, while $G$ and $D$ are trained by the original GAN loss objectives (Salimans et al., 2016). The hyperparameter $\lambda = 1$. In relatively complicated dataset such as SVHN, we add an auto-encoder term $\mathbb{E}_{\mathbf{x} \sim P_{\mathbf{x}}} \|G(E(\mathbf{x})) - \mathbf{x}\|^2$ in the objective function for only the last 1% training steps to further improve the reconstruction quality.

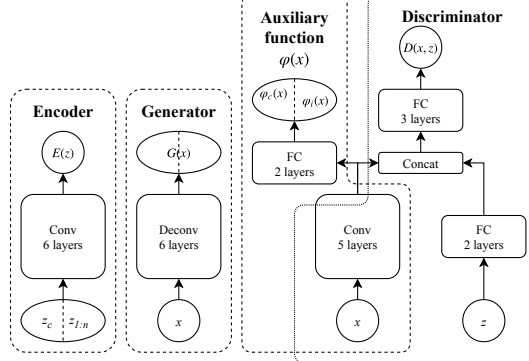

Figure 2: **Implementation** The encoder $E(x)$ is a convolutional network and the generator $G(z)$ is a deconvolutional network. The discriminator $D(x, z)$ shares parameters with the auxiliary function $\varphi(x)$. $\mathbf{z} = (z_c, z_{1:n})$ stands for the categorical and continuous codes.

## 4 EXPERIMENTS

We present our results in two parts: (1) Representing capability of semantic codes. We can store the information of an image by a few number of semantic codes, and the reconstruction from the codes keep the main features as the original one. (2) Defenses against gray-box and white-box attacks. In this paper, we call *gray-box* attacks as having access only to the original classifier architectures and parameters; *white-box* attacks are those have access to both of the classifier and FBGAN details.

We focus on three datasets in our experiments: the MNIST hand-written digits dataset (LeCun et al., 1998), Fashion MNIST (FMNIST) dataset (Xiao et al., 2017), and the Street View House Numbers (SVHN) dataset (Netzer et al., 2011).

### 4.1 SEMANTIC REPRESENTATION

FBGAN can present the semantic features of MNIST by one ten-dimensional categorical code and only four continuous codes, and FMNIST by one ten-dimensional categorical code and eight continuous codes. Previous related works require much higher latent space dimenssion. For example in InfoGAN, one ten-dimensional categorical code and three continuous codes and 128 random noises codes are used.

Categorical code can learn the most significant modes in a data distribution. For example, the ten-categorical code in MNIST / FMNIST represents ten different digits / fashion products. The

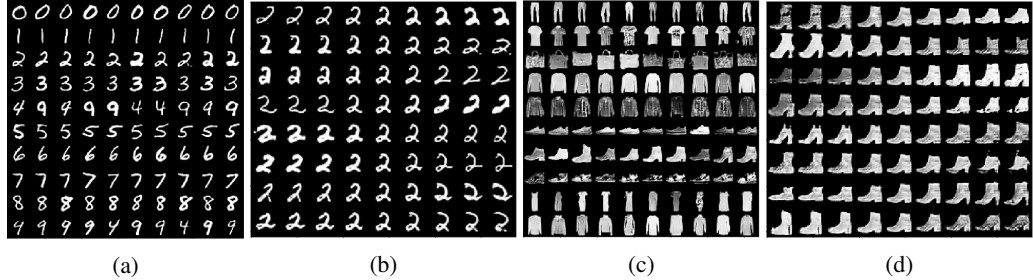

(a)      (b)      (c)      (d)

Figure 3: **Manipulating semantic codes on MNIST and FMNIST** Images generated by one ten-dimensional categorical code and eight continuous codes. (a) and (c) demonstrate that we can generate any category of images by changing the categorical codes. (b) and (d) are the effects of continuous codes: each row shows how the generated image changes when tuning one continuous codes with all other codes fixed.

Table 1: Classification accuracy (%) under different attack and defense methods for MNIST and FMNIST. The perturbation $\varepsilon$ is in $l_\infty$ norm. FBGAN here uses one ten-dimensional categorical code and 8 continuous codes. Gray-box attacks only apply to noise-filtering-type defense, and we compare FBGAN and Defense-GAN under the same setting. For white-box attack, the adversarial training with PGD $\varepsilon = 0.3$ is one of the state of the art results. Although better than FBGAN, adversarial training has its limitation: if the attack method is harder than the one used in training (PGD is harder than FGSM), or the perturbation is larger, then the defense may totally fail. FBGAN is effective and consistent for any given classifier, regardless of the attack method or perturbation.

| Attack | $\varepsilon$ | No defense | Gray-box | | White-box | | | |
| | | | FBGAN | Defense GAN | FBGAN | Adv train FGSM 0.3 | Adv train PGD 0.1 | Adv train PGD 0.3 |
|---|---|---|---|---|---|---|---|---|
| MNIST | | | | | | | | |
| Clean | 0 | 99.3 | 97.6 | 93.6 | 97.6 | 99.2 | 99.5 | 98.8 |
| FGSM | 0.1 | 78.2 | 96.6 | 95.2 | 93.4 | 97.4 | 97.9 | 97.6 |
| FGSM | 0.3 | 18.9 | 87.0 | 82.0 | 82.8 | 94.4 | 83.1 | 96.0 |
| PGD | 0.1 | 10.5 | 96.3 | 94.7 | 91.7 | 83.0 | 96.1 | 97.3 |
| PGD | 0.3 | 0.6 | 90.9 | 93.2 | 88.6 | 3.9 | 29.2 | 94.0 |
| FMNIST | | | | | | | | |
| Clean | 0 | 91.2 | 82.2 | 78.0 | 82.2 | 91.4 | 89.9 | 91.0 |
| FGSM | 0.1 | 24.2 | 76.3 | 52.6 | 62.7 | 82.6 | 81.0 | 75.9 |
| FGSM | 0.3 | 9.1 | 41.0 | 38.9 | 49.2 | 89.4 | 42.4 | 74.4 |
| PGD | 0.1 | 5.9 | 76.9 | 62.6 | 50.5 | 12.1 | 71.7 | 61.8 |
| PGD | 0.3 | 5.7 | 58.8 | 62.6 | 44.2 | 5.6 | 7.1 | 68.1 |

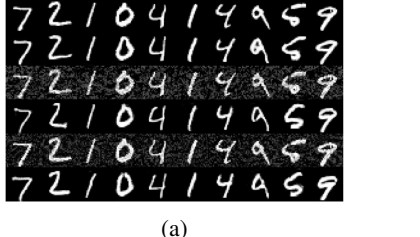
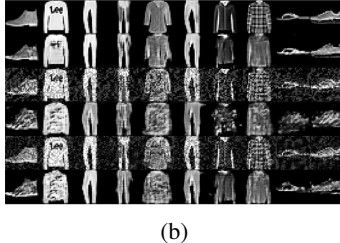

(a)                    (b)

Figure 4: **Reconstruction of MNIST and FMNIST**    The first two rows are the original test set images and their reconstructions; the middle two rows are the gray-box adversaries and their reconstructions; the last two rows are the white-box adversaries and their reconstructions. All the adversaries are from PDG with purtabation $\varepsilon = 0.3$.

continuous codes can finely tune the more detailed features of a certain mode. Figure 3 shows ten MNIST digits generated by FBGAN and the effect of tuning different continues codes.

We observe that the reconstruction of MNIST and FMNIST datasets are of high qualities. The encoder first encodes a semantic representation, which is then fed into the generator. The reconstructed image not only maintains the category, but also detailed features as the input.

## 4.2 ADVERSARIAL DEFENSES

### 4.2.1 DEFENSES AGAINST GRAY-BOX ATTACKS

In gray-box attacks, the attacker can only access to the classifier, but have no information about the FBGAN filter. Hence we prepare our adversarial data by using FGSM and PGD methods to directly attack trained classifiers. The classifier tested on the original MNIST dataset has accuracy of 99.26%, and the classifier tested on the original FMNIST dataset has accuracy of 91.16%. Table 1 shows our defense effect against different methods with different $\varepsilon$ values. As shown in Figure 4, given adversarial examples generated by PGD method with $\varepsilon = 0.3$, we have the reconstructed images with categories and main features maintained, and there are no more attack noises there.

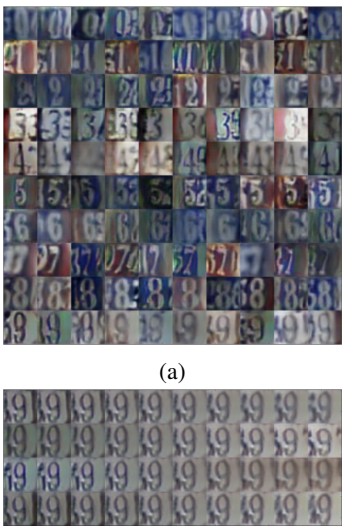

(a)

| Attack | $\varepsilon$ | No defense | FBGAN |
|--------|------|-----------|-------|
| Clean  | 0    | 93.7      | 83.4  |
| FGSM   | 0.05 | 11.4      | 66.4  |
| FGSM   | 0.10 | 10.8      | 47.7  |
| PGD    | 0.05 | 3.4       | 71.5  |
| PGD    | 0.10 | 2.9       | 60.9  |

(b)

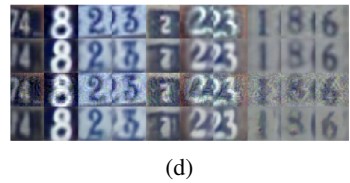

(c)                                            (d)

Figure 5: **Generation and reconstruction of SVHN**    (a) and (c) are generated images by changing the categorical codes and continuous codes respectively, similar to Figure 3. We observe that the continuous codes shown in (c) control: the blurriness (from clear to blurry), brightness (from bright to dark), background color (from green to brown) and the feature on the edge. (b) and (d) are the adversarial defense results. (b) shows the accuracy on clean, adversarial and reconstructed images, similar to Table 1. In (d), the first two rows are the clean images and their reconstructions, and the last two rows are the gray-box adversaries (PGD, $\varepsilon = 0.1$) and their reconstructions. The semantic codes consist 4 ten-categorical codes and 128 continuous codes.

### 4.2.2 DEFENSES AGAINST WHITE-BOX ATTACKS

In white-box case, the attacker can access not only the classifier but also the FBGAN filter. The original data $x$ is fed through the encoder $E$, the generator $G$ and the classifier $C$ to output $C(G(E(x)))$ as the classification. Since $E$, $G$ and $C$ are all represented as DNN, the whole structure is a large DNN and regraded as the objective of white-box attacks.

We implement white-box defense on MNIST and FMNIST with FBGAN having one ten-categorical code and eight continuous codes. A regularization is added to the encoded semantic codes $z = E(x)$: for the categorical code which is represented by a 10-dimensional probability vector, we replace it by the corresponding one-hot vector; for the continuous codes, we clip them between $[-1, 1]$. Regularizing the categorical codes can map the original input to its conterpart in the generated space, and clipping the continuous codes is to eliminate the influence of those low probability outliers. The results are shown in Figure 4 and Table 1, where the accuracy is above 82% on MNIST and 44% on FMNIST with adversarial perturbation $\varepsilon = 0.3$.

### 4.3 COMPARISON WITH BIGAN AND INFOGAN

BiGAN and InfoGAN are generative models aiming to produce new detailed data, while FBGAN is a defense model aiming to regenerate data with semantic features. The main novelty of FBGAN lies in combining the bidirectional mapping structure and feature extraction capability for the purpose of adversarial defense. The most important improvement from BiGAN and InfoGAN to FBGAN is the significant reduction of the number of semantic codes by applying MI regularization on all the semantic codes. BiGAN and InfoGAN require larger latent space to ensure the quality and diversity of the generation, and the semantic features are stored in latent codes in a highly entangled way; FBGAN requires much smaller latent space to catch the basic semantic features which is robust under attacks. For example, BiGAN and InfoGAN both employ at least 128 codes to represent and regenerate data of MNIST, while FBGAN reduces the number to 10 categorical codes and 4

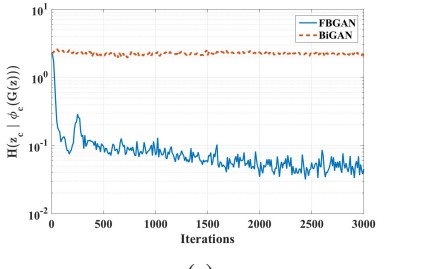 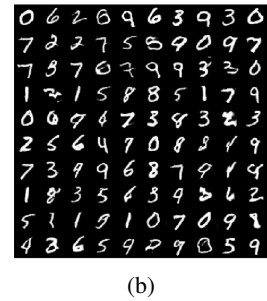 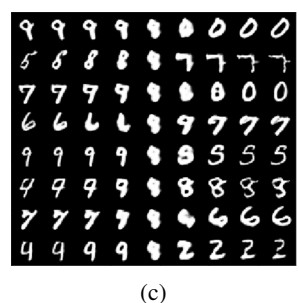

|(a)|(b)|(c)|

Figure 6: **Performance of vanilla BiGAN** (a) illustrates MI gap (5) of the categorical code, where FBGAN converges fast but BiGAN does not. (b) and (c) are generated images by changing the categorical code and continuous codes, similar to Figure 3. The semantic features are entangled in the latent codes of BiGAN.

continuous codes. Hence, generative models, such as BiGAN and InfoGAN, and FBGAN are tools for tasks in different domains.

Vanilla BiGAN without MI regularization cannot disentangle the semantic features. Theoretically, if BiGAN achieved its optimal solution, the minimization of JS divergence $D_{\mathrm{JS}}(P_{\mathbf{x},E(\mathbf{x})}\|P_{G(\mathbf{z}),\mathbf{z}})$ would ensure that $H(\mathbf{z}|G(\mathbf{z})) = 0$ and all latent codes are effective. However, experiments show that BiGAN cannot minimize the conditional cross entropy, and the latent codes cannot disentangle the semantic features automatically (Figure 6). Thus it is necessary to apply explicit MI regularization.

## 5 DISCUSSION

Nonetheless, the effectiveness of our FBGAN model against adversarial attacks are highly dependent on the reconstruction accuracy. It is also challenging to get a high reconstruction accuracy without over-fitting the training data. For example, in SVHN, we apply 4 ten-dimensional categorical codes and 128 continuous codes; however, its white-box defense accuracy is much worse than that of MNIST and FMNIST. We consider the various performances with different datasets as the fact that SVHN dataset has much more modes than the rest two datasets have. Even though the features within one category are quite different, for example different images of number one, the background of an image adds a large number of extra features to the object, which makes mode separation much harder. In contrast, MNIST and FMNIST dataset with all black background could be separated via fewer categorical codes. In our opinion, if we can find the suitable number of categorical codes, the performance of our model will be improved.

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
