# OpenReview forum: "Featurized Bidirectional GAN: Adversarial Defense via Adversarially Learned Semantic Inference"
_ICLR.cc/2019/Conference_

### Official Review · AnonReviewer2 · 2018-10-30
**Performs worse than adversarial training**

**Rating:** 3
**Confidence:** 4

**Review:**

This paper presents a new adversarial defense based on "cleaning" images using a round trip through a bidirectional gan.  Specifically, an image is cleaned by mapping it to latent space and back to image space using a bidirectional gan.  To encourage the bidirectional gan to focus on the semantic properties, and ignore the noise, the gan is trained to maximize the mutual information between z and x, similar to the info gan.

Pros:
	1. The paper presents a novel (as far as I am aware) way to defend against adversarial attacks by cleaning images using a round trip in a bidirectional gan

Cons:
	1. The method performs significantly worse than existing techniques, specifically adversarial training.
		a. The authors argue "Although better than FBGAN, adversarial training has its limitation: if the attack method is harder than the one used in training(PGD is harder than FGSM), or the perturbation is larger, then the defense may totally fail. FBGAN is effective and consistent for any given classifier, regardless of the attack method or perturbation."
		b. I do not buy their argument, however, because one can simply apply the strongest defense (PGD 0.3 in their results) and this outperforms their method in *all* attack scenarios.  And if someone comes out with a new stronger attack there's no guarantee their method will be strong defense against that method
	2. The paper is not written that well.  Even though the technique itself is very simple, I was unable to understand it from the introduction, and didn't really understand what they were doing until I reached the 4th page of the paper.


Missing citation:
PixelDefend: Leveraging Generative Models to Understand and Defend against Adversarial Examples  (ICLR 2018)

---

> ### Author Response · Authors · 2018-11-27
> **Reply**
>
> Thanks for your comments. And we really appreciate for your contribution and time. Here are our feedbacks to some of your concerns.
> 1. Weaker performance of FBGAN than that of adversarial training. Our argument is to emphasize that we do not need to re-train the classifier when we have adversarial examples with different attacks in FBGAN; however, adversarial training does need to re-train the classifier under different attacks. This is the limitation of adversarial training that we try to state. Actually, the defense mechanisms of our FBGAN and the adversarial training are quite different. Adversarial training improves the accuracy of the classifier by having access to as many adversarial examples with their corresponding correct labels as possible; while FBGAN only needs to train one classifier with the original clean data, and we may use this classifier to defend different attacks without any re-train process.
>
> 2. Both PixelDefend and our FBGAN are based on generative models, however, the mechanisms of utilizing generative models are quite different. PixelDefend reconstructs the images from adversarial examples pixel by pixel, which does not care about the overall structure or semantic meaning of images; FBGAN learns the semantic meanings of adversarial examples first, and use these semantic meanings to reconstruct images.
>
> Thanks again for your precious feedbacks.

---

### Official Review · AnonReviewer3 · 2018-10-31
**Novelty and evidence is not yet sufficiently clarified**

**Rating:** 4
**Confidence:** 5

**Review:**

This work proposes to defend against adversarial examples by “denoising” the input image through an autoencoder (a BiGAN trained similar to InfoGAN) before classifying it with a standard CNN. The robustness of the model is evaluated on the L_infinity metric against FGSM and PGD.

My main criticism is as follows:
* Novelty: several defences are based on a similar principle and the contributions of this paper are unclear.
* Insufficient evidence: The evaluation is minimal (only FGSM and PGD, no decision-, transfer- or score-based attacks) and insufficient to support the claims.
* Gradient masking: There is at least one clear sign of gradient masking in the results (FGSM performing better than PGD).

### Novelty
The only prior work against which the paper compares is DefenseGAN. The only advantage over DefenseGAN being stated is performance (because no intermediate optimisation step is used). However, besides DefenseGAN there are several other defences that project the input onto the learned manifold of “natural” inputs, including (see prior work section in [1] for an up-to-date list):

* Adversarial Perturbation Elimination GAN
* Robust Manifold Defense
* PixelDefend (autoregressive probabilistic model)
* MagNets

### Insufficient evidence
The only attacks employed are two gradient-based techniques (FGSM and PGD). It is known that gradient-based techniques may suffer from gradient-masking (see also next point) and that the effectiveness of different attacks various greatly (which is why one should use many different attacks). Hence, a full evaluation of the model should include score-based and decision-based attacks.

### Gradient masking
In Figure 5 (b) the FGSM attack performs better than PGD for epsilon = 0.05 (66.4% vs 71.5%). PGD, however, should be strictly more powerful than FGSM if the gradients and the hyperparameters are ok.

Gradient masking is the primary reason for why 95% of all proposed defences turned out to be ineffective, and there are good reasons to believe that the same might affect this defence. The robustness evaluation has to be much more thorough and convincing before any substantiated claims about the bidirectional architecture proposed here can be derived. In addition, the difference to prior work has to be made much clearer.

[1] Schott et al. “Towards the first adversarially robust neural network model on MNIST”

---

> ### Author Response · Authors · 2018-11-27
> **Reply**
>
> Thanks for your comments. And we really appreciate for your contribution and time. Here are our feedbacks to your concerns:
>
> 1. "Novelty: several defences are based on a similar principle and the contributions of this paper are unclear."
>
> At the time we submitted our paper, the only relevant defense mechanism we noticed at that time was DefenseGAN. There were also some other defense mechanisms which leveraged generative models but none of them attempted to extract semantic codes from the adversarial images, which is the main novelty of our model.
>
> Our contribuition is as following. FBGAN is the first model trying to understand the semantic meaning of an adversarial image and using this semantic meaning to reconstruct the original one. Our model is easily to be applied after training on the original data. On contrary, for example, defenseGAN, which also leverages the generative capability of GAN, needs to do search in the generated sample space every time it meets a new adversarial sample. FBGAN is not only faster but also has better performance than other generative model based defense methods.
>
> 2. "DefenseGAN is broken: the most similar work, DefenseGAN, has already been broken by Athalye et al. 2018, which is not discussed. The attacks deployed in this paper do not break DefenseGAN."
>
> We are very glad this reviewer mentioned the paper by Athalye et al. 2018. This work provided a very good method called BPDA which can defeat all seemingly strong methods related to so-called obfuscated gradient in last year’s ICLR. However, in that paper, they mentioned that defenseGAN was NOT broken at the time they wrote the paper. In addition, BPDA is an attack method to deal with those obfuscated gradient masking defend methods, which has nothing to do with DefenseGAN nor our FBGAN. Nonetheless, we are still happy to provide our defense result against BPDA method in that paper. Please see the second point in our reply to AnonReviewer1 for experiment details.
>
> 3. "Insufficient evidence: The evaluation is minimal (only FGSM and PGD, no decision-, transfer- or score-based attacks) and insufficient to support the claims."
>
> The evaluation methods used in our work are standard methods which are widely used in all other previous adversarial defense works. We don’t think the methods this reviewer mentioned are popular nor necessary to show the effectiveness of our work.
>
> 4. "Gradient masking: There is at least one clear sign of gradient masking in the results (FGSM performing better than PGD)."
>
> The reviewer believes that there exists gradient masking in Figure 5 (b). However, Figure 5 (b) is the results of gray-box attack, and the gray-box attack is calculated on the original non-robust classifier, so there is no gradient masking at all. Although the gradient masking may result in the fact that the defense accuracy of PGD is better than the defense accuracy of FGSM, it is not always true to claim that the gradient masking is the only reason that causes this phenomenon. Also our new experiment shows that BPDA, an attack method that works well on defenses utilizing the gradient masking, fails on our FBGAN (the detailed experiment results is shown under the feedback for AnonReviewer1).
>
> Thanks again for your feedbacks.

---

> > ### Comment · AnonReviewer3 · 2018-11-27
> > **some points taken, main concerns are not addressed**
> >
> > 2. "DefenseGAN is broken"
> > You are right about DefenseGAN not being broken.
> >
> > 3. The evaluation methods used in our work are standard methods which are widely used in all other previous adversarial defense works.
> >
> > FGSM and PGD are indeed widely used, but many previously proposed defences used additional attacks (like transfer-based, score-based, decision-based). Please check https://arxiv.org/pdf/1802.05666.pdf for an in-depth discussion of this issue.
> >
> > 4. "Gradient masking"
> > You are right that the gradient masking effects visible in the graybox attack doesn't necessarily indicate gradient masking in the white-box setting (but still means that hyper parameters of the attack have not been tuned properly).
> >
> > Given the discussion I will increase my score by one point, but the lack of a reliable robustness evaluation and the reduced novelty compared to DefenseGAN still puts it below the acceptance threshold in my opinion.

---

### Official Review · AnonReviewer1 · 2018-11-08

**Rating:** 3
**Confidence:** 4

**Review:**

Summary:
This paper gives a novel adversarial defense that consists of denoising images before classification. The denoising procedure consists of passing an image through a bidirectional GAN, which the authors use to map inputs to the latent space and then back to the original input space.

Novelty:
The exact mechanism through which this paper operates is novel, but many similar defenses have been proposed before that involve a latent space mapping followed by a mapping back to the original space; examples include DefenseGAN and PixelDefend.

Concerns:
- The evaluation is not thorough enough: Only two attacks are considered (FGSM and PGD, with the former being strictly weaker than the latter)
- DefenseGAN is similar in defense mechanism but the authors do not attempt to use the attacks of Athalye et al 2018 (ICML 2018) in their evaluation. We thus do not have strong lower bounds on adversarial robustness.
- In Figure 5b, the attack FGSM performs better than PGD, but FGSM is the single step case of PGD. This indicates that the attacks were not tuned properly, as you should always have PGD as a stronger attacker than FGSM
- The method does not perform as well as adversarial training in standard defense tasks
- Several writing/clarity errors (detailed below)

Smaller edits:
Page 2: paragraph 2: second last line: "feed" instead of "fed"
Page 2: bullet 1: under our contribution: line 3: "which are unchanged" instead of "which is unchanged"
Page 3: paragraph 3: second last line: "two distribution" missing an s (plural)
Page 3: Section 2.2: paragraph 2: line 2: "here are two most famous attacks" missing "the" before "two most famous"
Page 4: Section 3.2: first paragraph: line 4: "the latent codes is decomposed" should be "are" instead of "is"
Page 5: Paragraph 1: line 9: "E are trained" should be "E is trained"
Page 5: Section 4: Paragraph 1: last line: "are those have access " should be "are those which have access" missing which/that
Page 6: Last paragraph: Line 1: "the attacker can only access to the classifier" there is no need for "to"

---

> ### Author Response · Authors · 2018-11-27
> **Reply**
>
> Thanks for your comments. And we really appreciate for your contribution and time. Here are our feedbacks to your concerns:
>
> 1. "Only two attacks are considered (FGSM and PGD)."
>
> FGSM is the simplest and fastest adversarial attack method and it is widely used as a first-step robustness check for all state-of-art defense papers. As far as we know, most of attack methods invented by researchers in this community are based on the prototype FGSM, but with different approaches to doing gradient iteratively. Among those methods, PGD has been shown to be the strongest representative. Thus, our defense results under FGSM and PGD is enough to show our model's robustness.
>
> 2. "DefenseGAN is similar in defense mechanism but the authors do not attempt to use the attacks of Athalye et al 2018 (ICML 2018) in their evaluation. "
>
> AnonReviewer3 and his/her reference [1] claims that "Defense-GAN is broken" under the BPDA attack proposed by Athalye et al. (2018). However, Table 1 in Athalye et al. (2018) shows that Defense-GAN is one of the only two survivors under BPDA. We think that BPDA is an effective attack for gradient masking defense methods, but not for generative-model-based methods, so we did not test BPDA in our original paper.
>
> According to the reviewers' request, we implemented the following BPDA experiment: Recall that our prediction is C(G(E(x))), where E, G and C are the encoder, generator and classifier respectively. Following Section 4.1, 5.4 and Appendix B in BPDA, we approximate the backward pass of G(E(x)) with the identity function to calculate the adversarial images. For MNIST, the defense accuracy under Carlini and Wagner’s attack is 94.8% where the l_2 perturbation is 4.42, and the defense accuracy under PGD attack with l_\infty perturbation 0.3 is 91.6%. It suggests that FBGAN is robust under the attack aiming to gradient masking.
>
> 3. "In Figure 5b, the attack FGSM performs better than PGD, but FGSM is the single step case of PGD. This indicates that the attacks were not tuned properly, as you should always have PGD as a stronger attacker than FGSM."
>
> Frankly speaking, we don't quite understand the meaning of "the attacks were not tuned properly". The attack methods we used were all from CleverHans. We are pretty sure that we used them properly. In addition, methods like FGSM and PGD, the attack performance only depends on the bound of the perturbation. Furthermore, it is actually not necessary for PGD to always outperform FGSM in adversarial attack. PGD is a multi-step gradient-based method and FGSM is a single-step method. The performance of them depends on the landscape of the object function, which is still an unsolved question for deep learning community.
>
> 4. "The method does not perform as well as adversarial training in standard defense tasks."
>
> As mentioned in the main text, the defense performance of our FBGAN highly depends on the training of GAN. Adversarial training is a method only requires doing maximum worst case optimization during training process and it does not require extra networks' training. Thus, it is unfair to compare these two different mechanisms together. If comparing with method of the same category which also using generative model as a re-constructor, for example DefenseGAN, our model outperforms it by 1.4% , 5.0% on FGSM 0.1 and 0.3 attack respectively.
>
> Thanks again for bringing out those typos, we will correct all of them in our next revision.

---

### Meta-Review · Area_Chair1 · 2018-12-17
**Reject**

**Confidence:** 5
**Recommendation:** Reject

**Metareview:**

The reviewers agree the paper is not ready for publication.